# Accurate Bayesian phylogenetic point estimation using a tree distribution parameterized by clade probabilities

**Lars Berling** [ID][1,2], **Jonathan Klawitter** [ID][3], **Remco Bouckaert**[3], **Dong Xie**[3], **Alex Gavryushkin** [ID][1,2], **Alexei J. Drummond**[3]*

**1** School of Mathematics and Statistics, University of Canterbury, Aotearoa, New Zealand, **2** Biomathematics Research Centre, University of Canterbury, Aotearoa, New Zealand, **3** Centre for Computational Evolution, University of Auckland, Aotearoa, New Zealand

* a.drummond@auckland.ac.nz

**Data availability statement:** The CCD-MAP trees present a significant advancement over

## Abstract

Bayesian phylogenetic analysis with MCMC algorithms generates an estimate of the posterior distribution of phylogenetic trees in the form of a sample of phylogenetic trees and related parameters. The high dimensionality and non-Euclidean nature of tree space complicates summarizing the central tendency and variance of the posterior distribution in tree space. Here we introduce a new tractable tree distribution and associated point estimator that can be constructed from a posterior sample of trees. Through simulation studies we show that this point estimator performs at least as well and often better than standard methods of producing Bayesian posterior summary trees. We also show that the method of summary that performs best depends on the sample size and dimensionality of the problem in non-trivial ways.

## Author summary

Our research introduces novel methods to analyse a set of phylogenetic tree topologies, such as those generated by Bayesian Markov Chain Monte Carlo algorithms. We define a new model for a distribution on trees that is based on observed clade frequencies. We study it together with closely related models that are based on observed clade split frequencies. These distributions are easy to work with and, as we show experimentally, provide excellent estimates of the true posterior distribution. Furthermore, we demonstrate that they enable us to find the tree with the highest posterior probability, which acts as a summary tree or point estimate of the distribution. In simulation studies, we show that the new methods performs as least as well or better than existing methods. Additionally, we highlight that choosing the best method for summarizing sets of trees remains challenging, as it depends on the sample size and complexity of the problem in non-trivial ways. This work has the potential to improve the accuracy of phylogenetic studies.

the current standard method (MCC) for point estimators in BEAST1/2, offering fast, efficient, and notably improved performance. These new point estimators are freely available at https://github.com/CompEvol/CCD/ as a package for BEAST2. The installation instructions and user manual are also provided at the GitHub. The Accurate Bayesian phylogenetic point estimation using a tree distribution 23 simulated datasets and DS1 to DS4 used for the evaluation are also freely available under doi: 10.17608/k6.auckland.c.7102354.

**Funding:** LB, JK, AG, and AJD were partially supported by the Beyond Prediction Data Science Research Programme (MBIE grant UOAX1932). AG was partially supported by a Marsden Grant from the Royal Society Te Aparangi 21-UOC-057. The funders had no role in study design, data collection and analysis, decision to publish, or preparation of the manuscript.

**Competing interests:** The authors have declared that no competing interests exist.

## Introduction

One of the main inference paradigms in phylogenetics is Bayesian inference using Markov Chain Monte Carlo (MCMC)[1–3]. The distinguishing characteristic of phylogenetic models is the tree topology describing the evolutionary relationships for a set of taxa. Bayesian inference is based on a statistical model that describes the probability of a set of sequences given a phylogenetic tree, consisting of a topology with associated divergence times (or branch lengths) and model parameters. The MCMC algorithm iteratively samples a state space that, if set up with appropriate length and sampling interval, returns a sample that is a representation of the true underlying posterior distribution. In the case of phylogenetic MCMC algorithms, the output of such an analysis is a sample of phylogenetic trees, typically numbering in the thousands.

In a Bayesian phylogenetic analysis, the posterior distributions of many continuous parameters (e.g. kappa, base frequencies, molecular clock rate, population size) are easily summarised by considering statistics of the marginal distribution of the parameter of interest from the samples obtained by MCMC. On the other hand, one of the most crucial parameters—the tree topology—is a discrete parameter whose central tendency and variance are harder to characterise due to the high-dimensional and non-Euclidean nature of tree space [4–6]. It has thus become standard practise to employ summary or consensus tree methods to condense the output into a single tree [7]. Although we focus on Bayesian phylogenetics in this paper, it is worth noting that this approach is not unique to it but rather commonly employed across the field of phylogenetics when analysing collections of trees. This single tree, which in this paper we refer to as a Bayesian *point estimate*, is then used for further representation and interpretation of an analysis. Despite considerable efforts dedicated to the development of summary methods [8], it remains unclear which method performs best for summarising collections of trees. Most summary methods construct a tree in two steps [7]: First, a tree topology is constructed or selected, and, second, this discrete topology is then annotated with divergence times (or branch lengths). In this paper we focus on the first step, the construction of a rooted binary tree topology.

The predominant challenge for many summary tree estimators is the complexity of the tree space they are operating on. This is particularly the case for methods trying to compute a mean in a high-dimensional, non-Euclidean space such as the Billera-Holmes-Vogtmann (BHV) space [5,9,10] or a space induced by rearrangement operations [6]. While good progress has been made, these methods suffer from the complexity of tree space geometry and are not tractable yet for large problems [6,10]. The two most popular methods in practice thus operate only on the sampled trees. First, consensus methods focus on finding a consensus among the given trees. The prevalent variant is the *greedy majority-rule consensus (greedy consensus or MRC) tree*, which builds up a tree by including clade after clade greedily (i.e., more frequent clades first) that are compatible with the current tree; ties are broken arbitrarily [8]. Consensus methods are however prone to polytomies (i.e., parts of the tree remain unresolved) and finding the most resolved greedy MRC tree is an NP-hard problem [11]. Second, the *maximum clade credibility (MCC) tree* picks the tree from the sample distribution with maximum product of (Monte Carlo) clade probabilities. While the computation of the MCC tree is fast and efficient, it comes at a cost in accuracy due to the restriction to the sampled trees. The equivalent of this tree outside the BEAST framework would be the sampled tree with highest posterior probability, commonly used within MrBayes [2].

A good estimate of the tree distribution is still needed for questions concerning, for example, the credibility set of trees and the information content (entropy) [12] as well as for

applications such as Bayesian concordance analysis (BCA) [13]. Introduced by Höhna and Drummond [14] and improved by Larget [15], the *conditional clade distribution (CCD)* offers an advanced estimate of the posterior probability distribution of tree space. Based on simple statistics of the sample, it provides normalized probabilities of all represented trees and allows direct sampling from the distribution. CCDs have for example been used to measure the information content and detect conflict among data partitions [12], for species tree–gene tree reconciliation [16], and for guiding tree proposals for MCMC runs [14]. Constructing the CCD and performing these tasks can be done efficiently [12,15]. Zhang and Matsen [17,18] and Jun et al. [19] looked at a slightly more complex model than a CCD, called a subsplit directed acyclic graph (sDAG). While similar to a CCD, probabilities in an sDAG can be obtained from a sample of trees, they also discuss different methods to learn the model parameters [17–19].

In this paper we extend the applicability of CCDs by introducing a new parametrization for CCDs and describing fixed-parameter tractable algorithms to compute the tree with highest probability. We demonstrate the usefulness of the new distribution and these new point estimates for Bayesian phylogenetics by comparing them to existing methods in simulation studies. Particularly, we find that these point estimates generally outperform the MCC tree and are more robust to the random sampling process of MCMC.

## Methods

In this section, we first discuss properties of tractable tree distributions and define CCDs with three different parametrizations. We then recall the definitions of the MCC and greedy consensus tree and show how CCDs give rise to new point estimators. Lastly, we describe the datasets we generated for our experiments. Throughout, we write tree instead of tree topology and further assume that all our trees are rooted and, unless mentioned otherwise, are binary.

### Tractable tree distributions

The following are some key criteria we would like any distribution over a set of trees to meet. While not a formal definition, these are important desiderata for any such distribution to satisfy.

We consider a probability distribution over a set of trees (on the same taxa) a *tractable tree distribution* if some common tasks can be performed efficiently in practice. Example tasks are computing the probability of a tree and retrieving the tree with maximum probability. As the main quality criteria for a tractable tree distribution we consider its *accuracy*, that is, how well it estimates the probability of trees, in particular of those in the 95% credibility set. In simulation studies we can also test whether a distribution contains the true tree. If we generate a type of distribution for the same data multiple times, we can consider the *precision* and the *stability*, that is, how much the probabilities of trees and how much the accuracy change, respectively. Since below we populate the parameters of CCDs deterministically from samples, we can only measure these indirectly through samples from different MCMC runs.

A simple example distribution is the set of sampled trees from an MCMC run; we call this a *sample distribution*. It offers Monte Carlo probabilities and while some tasks can be performed efficiently, it has quite low accuracy, poor representativeness, and is in general not stable. In fact, since the space of trees increases super-exponentially with the number of taxa, a sample on several thousand trees typically misses the majority of trees with non-negligible posterior probability even for moderate size problems.

Reintroducing the concept of a CCD, we first define a graph, which we call a *forest network*, capable of representing a larger number of trees. Assigning probabilities to certain vertices (or edges), we obtain a *CCD graph*. The version of a CCD by Larget [15] is one possible parametrization of a CCD based on observed clade splits; we call this a CCD1. Our new parametrization, CCD0, is based on observed clades. Here we use the observed clade split and clade frequencies to populate these parameters. We also show how to efficiently sample trees from a CCD and how dynamic programming allows efficient computation of values such as the number of trees and its entropy.

**Forest network.**  Let $X$ be a set of $n$ taxa. A *forest network $N$* on $X$ is a rooted bipartite digraph with vertex set $(\mathcal{C}, \mathcal{S})$ that satisfies the following properties:

- Each $C \in \mathcal{C}$ represents a *clade* on $X$. So for each $C \in \mathcal{C}$, we have $C \subseteq X$; for each taxon $\ell \in X$, $\{\ell\} \in \mathcal{C}$, and also $X \in \mathcal{C}$.
- Each $S \in \mathcal{S}$ represents a *clade split* (also called *subsplit* in the context of unrooted trees [17]). So each $S \in \mathcal{S}$ has degree three with one incoming edge $(C, S)$ and two outgoing edges $(S, C_1)$, $(S, C_2)$ such that $C_1 \cup C_2 = C$, $C_1 \cap C_2 = \emptyset$ for some $C_1, C_2, C \in \mathcal{C}$. Then $S = \{C_1, C_2\}$ and $S$ is a clade split of $C$. We also use the notation $C_1 \| C_2$ for $S$.
- Each non-leaf clade has outdegree at least one and each clade except $X$ has indegree at least one.

Note that $X$ is the root of $N$, the taxa in $X$ are the leaves of $N$, and each non-leaf clade has at least one clade split. We use terms such as *child* and *parent* naturally to refer to relations between vertices of $N$. (For example, the root clade in the forest network in Fig 1B has three child clade splits and each clade split $S$ has a parent clade $C$.) When talking about multiple graphs, we let $\mathcal{C}(N)$ and $\mathcal{S}(N)$ denote the clades and clade splits, respectively, of $N$. For a (rooted binary phylogenetic) tree $T$ on $X$, we use analogous definitions for $\mathcal{C}(T)$ and $\mathcal{S}(T)$ (each pair of sibling clades in $T$ forms a clade split of $T$). For a clade $C$, we define $\mathcal{S}(C)$ as the set of child clade splits of $C$.

A forest network $N$ *displays* or *contains* a tree $T$ if each clade split of $T$ is in $\mathcal{S}(N)$, i.e., $\mathcal{S}(T) \subseteq \mathcal{S}(N)$; see Fig 1 For a clade $C$ define $N(C)$ as the restriction of $N$ to $C$, that is, the forest subnetwork rooted at $C$ containing all vertices reachable from $C$. Analogously, for $S \in \mathcal{C}(N)$, we can define the forest subnetwork $N(S)$ of $N$ that is rooted at the parent clade $C$ of $S$ but contains only $S$ as child of $C$ and all vertices reachable from $S$. Note that, for a clade split $\{C_1, C_2\}$ of $X$, network $N$ contains all trees composed (amalgamated) of one subtree from $N(C_1)$ and one subtree from $N(C_2)$; this holds recursively. Hence, a forest network is suitable to represent huge numbers of trees when all combinations of subtrees are included.

**CCD graph.**  In order to turn a forest network into a tree distribution, we need to be able to compute a probability for a tree $T$. Larget [15] suggested to use the product of clade split probabilities over all clade splits in $\mathcal{S}(T)$ as the probability of $T$. We define a *CCD graph* as a forest network $G$ where each clade split $S$ in $\mathcal{S}(G)$ has an assigned probability $\Pr(S)$ such that, for each clade $C \in \mathcal{C}(G)$, we have $\sum_{S \in \mathcal{S}(C)} \Pr(S) = 1$; see again Fig 1B. In other words, we can randomly pick a clade split at $C$. From Larget [15,Appendix 2] we then get that $G$ represents a tree distribution. So for a tree $T$ displayed by $G$, we have

$$\Pr(T) = \prod_{S \in \mathcal{S}(T)} \Pr(S) \tag{1}$$

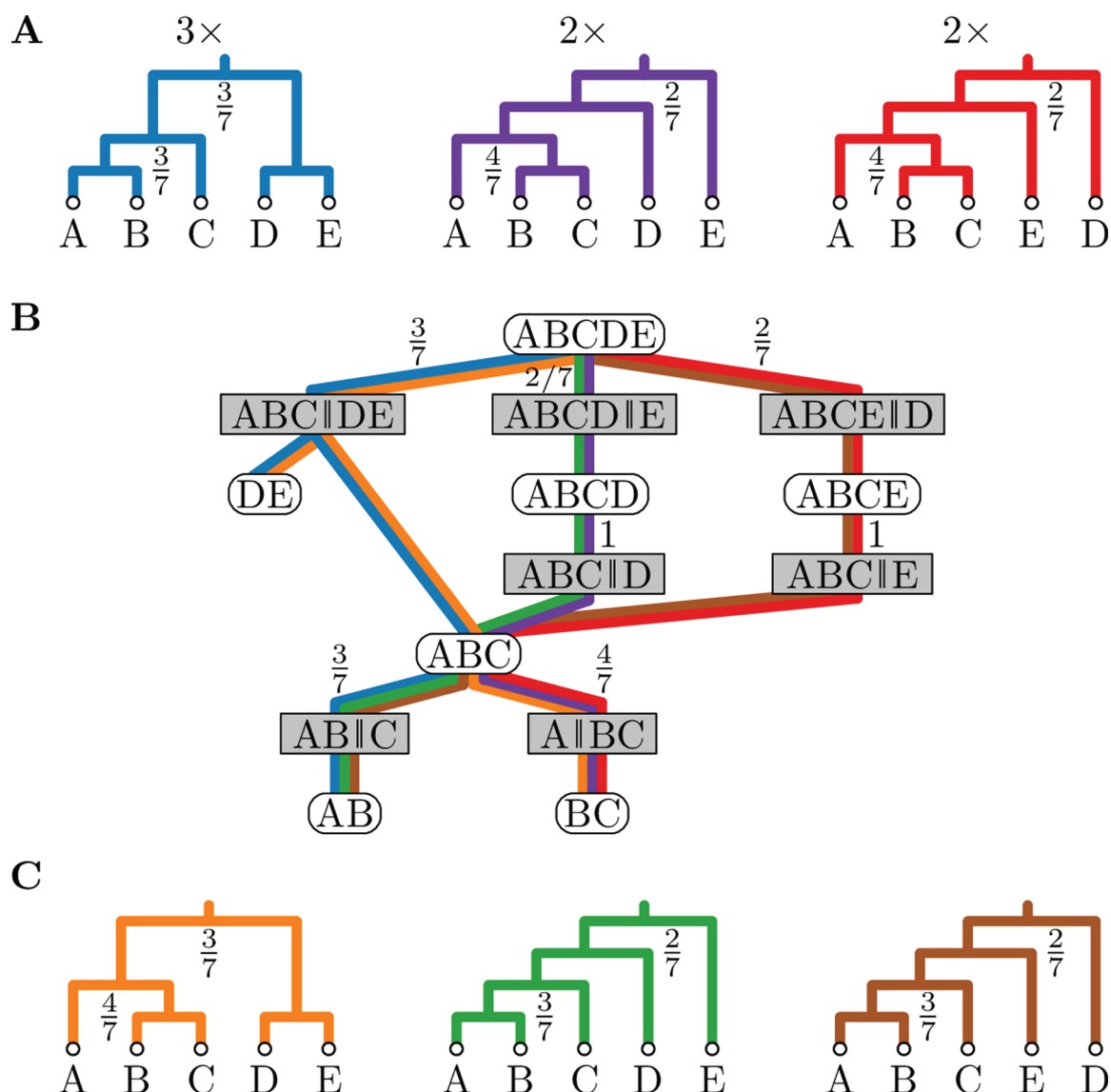

**Fig 1. A CCD graph** ((B) forest network with clade split probabilities) based on a tree sample (A) smoothens the probabilities to all trees it displays: (A) Posterior sample of size seven consisting of three different trees sampled thrice, twice, and twice. Only the clades ABCDE and ABC are split in multiple ways. The resulting probabilities of the trees in the CCD1 are thus 9/49, 8/49, and 8/49. (B) Truncated CCD graph (cherry splits and singletons omitted) based on the sample trees above also displays the unsampled trees below. (C) Unsampled trees with CCD1 probabilities 12/49, 6/49, and 6/49, respectively.

and, for any other tree $T'$, we have $\Pr(T') = 0$. Furthermore, the sum of probabilities of all trees displayed by $G$ is one. We now show how CCD1 and CCD0 assign probabilities based on observed clade split and clade frequencies, respectively.

**CCD1, observed clade splits.** CCD1 is a tree distribution over the space of trees on a fixed set of taxa $X$ based on a CCD graph with clade split probabilities obtained as follows. Let $\mathcal{T} = \{T_1, \ldots, T_k\}$, a (multi-)set of trees on $X$, e.g., the samples of an MCMC run. Let $\mathcal{C}$ and $\mathcal{S}$ be the sets of clades and clade splits appearing in $\mathcal{T}$, respectively. Then let $G$ be the forest network induced by $\mathcal{T}$, that is, $G$ has vertex set $\mathcal{C} \cup \mathcal{S}$ and edges naturally induced by the

clade splits $\mathcal{S}$ (we know the two child clades and the parent clade of each clade split). Furthermore, we assign clade split probabilities as follows to turn $G$ into a CCD graph. For a clade $C \in \mathcal{C}$ and a clade split $S \in \mathcal{S}$, let $f(C)$ and $f(S)$ denote the frequencies of $C$ and $S$ appearing in the sample $\mathcal{T}$, respectively. Note that

1. $f(S) \leq f(C)$ for all pairs of $S, C$ with $S \in \mathcal{S}(C)$;
2. $\sum_{S \in \mathcal{S}(C)} f(S) = f(C)$ for each non-leaf clade $C$;
3. $f(X) = k$ and, for each $\ell \in X, f(\{\ell\}) = k$.

The *conditional clade probability (CCP)* $\Pr(S)$ of a clade split $S$ is defined as the ratio of $S$ being the split of $C$ in the posterior sample, i.e.,

$$\Pr(S) = f(S)/f(C) \qquad (2)$$

Note that $\sum_{S \in \mathcal{S}(C)} \Pr(S) = 1$ and $\Pr(S) = 1$ if $S \in \mathcal{S}(\{a, b\})$ for some leaves $a, b$. The resulting CCD graph is what we call a *CCD1*, the conditional clade distribution induced by the probability distributions of clade splits.

**Example.** Let us consider the example shown in Fig 1 where the posterior samples consists of three trees with the first being sampled three times, and the others twice each. Observe that the root clade ABCDE is split in three different ways, namely, ABC‖DE, ABCD‖E, and ABCE‖D. The probabilities of these three clades splits are $\Pr(ABC\|DE) = 3/7$, $\Pr(ABCD\|E) = 2/7$, and $\Pr(ABCE\|D) = 2/7$. Furthermore, the clade ABC is split in two different ways with probabilities $\Pr(AB\|C) = 3/7$ and $\Pr(A\|BC) = 4/7$. All other clades are trivial or are only split in one way, e.g., the clade ABCD is always split into ABC‖D, so $\Pr(ABC\|D) = 1/1$.

The resulting CCD contains 6 different trees – the three sampled trees as well as three unsampled trees (Fig 1C). Note that the tree sampled most often still has the highest probability, with $3/7 \cdot 3/7 = 9/49$, among the sampled trees, as the other two trees have a probability of $1/7 \cdot 4/7 = 4/49$ each. Furthermore, the unsampled tree containing the most frequent clade split ABC‖DE of the root clade and the most frequent clade split A‖BC of ABC, has a higher probability of $3/7 \cdot 4/7 = 12/49$.

**CCD0, observed clades.** For the new CCD0, our goal is to have a distribution where the probability of a tree is based on the product of its clades' frequencies. We could derive the probability of a clade $C$ from a posterior sample on $k$ trees as $\Pr'(C) = f(C)/k$. While in general this does not yield a distribution as the tree probabilities do not sum to one, we can compute the normalizing factor; in fact, we can even compute the normalizing factor per clade split. Since for complex problems even large samples may not contain all plausible clade splits, we have as another feature for CCD0 that we also include (some) non-observed clade splits.

A CCD0 is again based on a forest network $G$ with clades $\mathcal{C}$ as before as those appearing in $\mathcal{T}$ and the clade splits $\mathcal{S}$ defined as follows. Let $\mathcal{S}$ be the set of all possible clade splits that can be formed from $\mathcal{C}$, that is, for any three clades $C_1, C_2, C \in \mathcal{C}$ with $C_1 \cup C_2 = C$ and $C_1 \cap C_2 = \emptyset$, we have $\{C_1, C_2\} \in \mathcal{S}$. (In the example above, there are no additional clade splits besides the observed ones for CCD1.) We turn $G$ into a CCD graph by turning the clade frequencies into clade split probabilities (with an algorithm explained in Sect S1.1 of S1 Text). In particular, the clade split probabilities are set such that the probability of any tree $T$ in $G$ given by Eq (1) is equal to product of (Monte Carlo) clade probabilities, $\Pr'(T) = \prod_{C \in \mathcal{C}(T)} f(C)/k$, *normalized* over all trees in $G$.

Both CCD0 and CCD1 are estimates of the true posterior tree distribution. Their models assume that clades/clade splits in one part of a tree behave independent of other clades. So

a CCD smoothens the probabilities of a sample distribution by moving probability of over-represented sampled trees to trees that have not been sampled, but whose clades/clade splits appear within the samples. CCD0 provides a simpler model since it is only based on observed clades whereas a CCD1 is based on clade splits and thus has more parameters. Here we use the observed frequencies to populate these parameters, but other methods such as Maximum Likelihood optimization and variational methods could be investigated [18,19].

**Example, continued.** Note that the three sampled trees from Fig 1A result in the same CCD graph for CCD0 and CCD1 as no potential pair of a child clades can be combined into an unobserved parent clade. In contrast, in the example in Fig 2, CCD0 and CCD1 are different as CCD0 contains the clade split AB∥CD (we observe clades AB, CD, and ABCD) but CCD1 does not (we do not observe this clade split).

**CCD2 and further tree distributions.** Similar to a CCD graph, Zhang and Matsen [17] and Jun et al. [19] use a structure they call a *subsplit directed acyclic graph (sDAG)*. Here the vertices are clade splits as well as a root clade and leaf clades with an edge when a clade/both clades of a clade split correspond to one clade of the parent clade split. They then add probabilities to the edges to turn it into a distribution. An sDAG thus describes a model where the probability of a clade split $\{C_1, C_2\}$ not only depends on its parent clade $C$ but also the clade split $C$ is part of. This model has thus more parameters and is more complex than CCD1 and CCD0. Furthermore, we can represent the core structure of this model, which we call *CCD2*, with an *extended CCD graph* where each clade vertex is further distinguished by sibling clades; see Fig 3 for the extended CCD graph of the example in Fig 1. In this paper we focus on populating the CCD2 parameters solely using the tree sample, whereas Zhang and Matsen [17,18] studied other approaches to compute sDAGs and its parameters, applying more advanced techniques such as regularization and variational methods.

Dumm et al. [20] extended an sDAG to a *history sDAG* by adding labels (e.g. ancestral sequences) to each vertex, so that a clade/clade split can exist multiple times but with different labels. They use history sDAGs to represent and find maximally parsimonious trees.

*Remark.* When computing a CCD1 or CCD2 based on a tree sample $\mathcal{T}$, it is important that $\mathcal{T}$ does not contain outlier trees that should have been discarded as burnin. Suppose otherwise, that there is an outlier tree $T$ that does not share any clades (except $X$ and the taxa) with the other trees in $\mathcal{T}$. Then $X$ has one clade split $S_T$ corresponding to $T$ with $\Pr(S_T) = \frac{1}{k}$; all other non-leaf clades of $T$ have only one clade split and so probability 1. Therefore, $\Pr(T) = \frac{1}{k}$ and is thus vastly overestimated. However, it is possible to built a simple heuristic to detect such outliers: For example, one could check if removing a tree $T$ from the CCD and thus

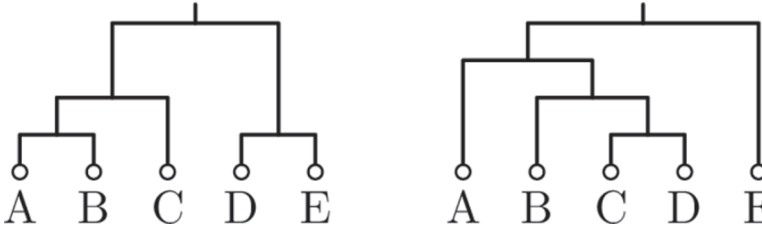

**Fig 2. For this sample of trees, the CCD graph of CCD0 and CCD1 differ since AB and CD can form an unobserved clade split.**

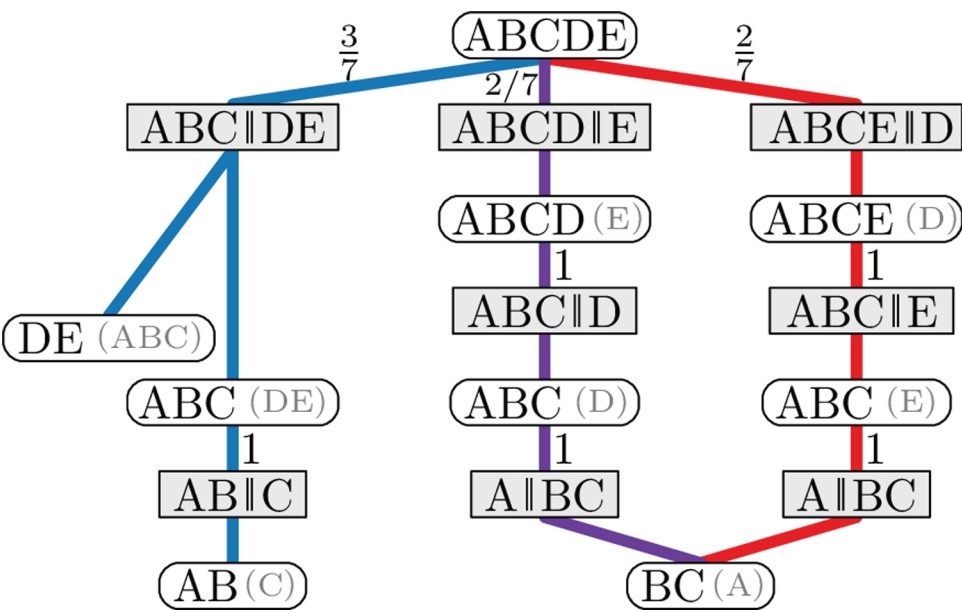

**Fig 3. Truncated extended CCD graph (cherry splits and singletons omitted, sibling clades in brackets) based on the sample trees from Fig 1A that represents a CCD2.** Note that it only contains the three sampled trees. (While the clade vertices might seem redundant here, they have in general higher in- and outdegree.)

decreasing clade and clade split frequencies by one, significantly change the probability of $T$. Another option is too check if $T$ contains any clades or clade splits that have been observed only once. Nonetheless, this behaviour should be kept in mind when working with CCD1 and CCD2, and in particular when $\mathcal{T}$ contains only few different trees.

**Utilizing CCDs.** With the CCD graph as data structure underlying CCD0, CCD1, and CCD2, we can efficiently sample and compute interesting values over a whole CCD. To sample a tree from a CCD, starting at the root clade $X$, pick a clade split $\{C_1, C_2\}$ among $\mathcal{S}(X)$ based on their probabilities; then proceed in the same fashion with $C_1$ and $C_2$ until a fully resolved tree is obtained.

We can also use dynamic programming to compute values such as the number of different trees (topologies) and the entropy of a CCD, or (as explained below) find the tree with maximum probability. For example, to compute the number of different trees in a CCD graph $G$, for a clade $C$, let $t(C)$ be the number of different trees in $G(C)$. For a leaf $\ell$, we have $t(\ell) = 1$, and for any other clade, we can use the following recursive formula:

$$t(C) = \sum_{\{C_1, C_2\} \in \mathcal{S}(C)} t(C_1)t(C_2) \tag{3}$$

Using dynamic programming, we compute these values bottom-up through $G$. Then $t(X)$ is the total number of different trees in $G$. Note that this calculation takes linear time in the number of clades and clade splits.

Analogously, we can compute the entropy of the CCD by computing, for each clade $C$, the entropy of $G(C)$; let $H^\star(C)$ denote this value. We can then use the formula by Lewis et al. [12]:

$$H^\star(C) = \sum_{\substack{S \in \mathcal{S}(C) \\ S = \{C_1, C_2\}}} -\Pr(S)\left(\log \Pr(S) - H^\star(C_1) - H^\star(C_2)\right) \tag{4}$$

where for each leaf $\ell \in X$ we have $H^\star(\ell) = 0$. The entropy of the CCD is then $H = H^\star(X)$. Note that $\exp(-H)$ is the average probability of a tree in the CCD and we can define $N_e = \exp(H)$ as the *number equivalent* – the effective number of distinct topologies in the distribution.

## Point estimators

We recall the definitions of the two most commonly used point estimators and define new point estimators based on CCD0 and CCD1. Let $\mathcal{T}$ be again a tree sample on $k$ trees for which we can compute the frequencies for trees, clades, and clade splits.

**MCC tree.** Let $\Pr_{CC}(C)$ denote the clade credibility (Monte Carlo probability) of clade $C$, i.e., $\Pr_{CC}(C) = f(C)/k$. The *clade credibility* $\Pr_{CC}(T)$ of a tree $T \in \mathcal{T}$ is the product of its clades' clade credibilities:

$$\Pr_{CC}(T) = \prod_{C \in \mathcal{C}(T)} \Pr_{CC}(C) \tag{5}$$

The *maximum clade credibility (MCC) tree* is the tree $T$ in $\mathcal{T}$ that maximizes $\Pr_{CC}(T)$. Note that the MCC tree is restricted to be from the sample.

**Greedy consensus tree.** Let $C_1, \ldots, C_m$ be the nontrivial clades appearing in $\mathcal{T}$ ordered by decreasing frequency; ties are broken arbitrarily. Starting with a star tree $T'$ with root $X$ and leaves $\{\ell\}$, $\ell \in X$, we process the clades in order. For the next clade $C_i$, we test whether $C_i$ is compatible with current tree $T'$, that is, whether there is a clade (vertex) $C$ containing $C_i$ in $T'$ and with no child clade $C'$ of $C$ containing or properly intersecting $C_i$. If we find such a clade $C$, we refine $T'$ by making $C_i$ a new child of $C$ and making all child clades of $C$ that are contained in $C_i$ child clades of $C_i$. After $C_m$, the resulting tree is the *greedy consensus tree*. For $n$ taxa and $k$ trees, the greedy consensus tree can be computed in $\mathcal{O}(k^2 n)$ time or $\mathcal{O}(kn^{1.5} \log n)$ time [11,21], or in $\widetilde{\mathcal{O}}(nk)$ time [22] ($\widetilde{\mathcal{O}}(\cdot)$ ignores logarithmic factors).

**CCD-based point estimators.** For a CCD[$i$], $i \in \{0, 1, 2\}$, we call the tree $T$ with maximum probability $\Pr(T)$ in the CCD[$i$] the *CCD[i]-MAP tree*. Using the recursive relationships for CCDs explained above, we can find the CCD[$i$]-MAP tree efficiently as follows. Let $\Pr^\star(C)$ denote the maximum probability of any subtree rooted at clade $C$. With $\Pr^\star(\ell) = 1$ for a leaf $\ell$, we can compute $\Pr^\star(C)$ with the following formula:

$$\Pr^\star(C) = \max_{\{C_1, C_2\} \in \mathcal{S}(C)} \left\{ \quad \Pr(C_1, C_2 \mid C) \cdot \Pr^\star(C_1) \cdot \Pr^\star(C_2) \quad \right\} \tag{6}$$

The maximum probability of any tree in the CCD[$i$] is then given by $\Pr^\star(X)$. The tree $T$ achieving this maximum probability can be obtained along with the corresponding value by dynamic programming.

Note that the CCD0-MAP tree is based on the same criteria as the MCC tree since the clade split probabilities in a CCD0 are based on clade frequencies. However, the choice for

the CCD0-MAP tree is not restricted to the sample. Further note that the greedy consensus greedily picks clades based on their clade credibility. We combine these two ideas into another point estimator for CCD0. The *CCD0-MSCC tree* ('S' for 'sum') is the tree in the CCD0 that maximizes the sum of clade credibilities.

When annotating a tree $T$ obtained with a CCD with clade support, an alternative to the Monte Carlo probabilities from the MCMC run is to use the probability of each clade of $T$ to appear in a tree of the CCD. In fact, the probability of a clade $C$ in a CCD1 equals the Monte Carlo probability of $C$ in the sample used to construct the CCD1 (see Sect S1.1 of S1 Text for a proof). For a CCD0 or if the parameters were set differently, clade probabilities can be computed efficiently with the CCD graph.

## Datasets

We performed well-calibrated simulation studies [23] using the LinguaPhylo packages LPhyStudio and LPhyBEAST [24] and BEAST2 [1] to obtain posterior samples. We used both Yule tree and time-stamped coalescent simulations. (See Figs A and B of S1 Text for graphical models.)

For our Yule tree simulations we generated two sets of 250 trees and alignments with 10 and 20 ($n$) taxa, as well as 100 trees and alignments with 50, 100, 200 and 400 taxa. For all simulations (except $n = 20$) the birth rate of the Yule [25] process was fixed to 25.0 (12.5 for $n = 20$). For the substitution model, we used the HKY+G model [26]. The shape parameter for the gamma distribution of site rates was modelled using a log-normal distribution, with a mean in log space of –1.0 and a standard deviation in log space of 0.5. The transition/transversion rate ratio ($\kappa$) also followed a log-normal distribution, with a mean in log space of 1.0 and a standard deviation in log space of 1.25. The nucleotide base frequencies were independently simulated for each replicate from a Dirichlet distribution with a concentration parameter array of [5.0, 5.0, 5.0, 5.0]. The length of the sequence alignments was 300 sites (600 sites for $n = 20$) and the mutation rate was fixed at 1.0, so that divergence ages were in units of substitutions per site. In addition, we generated another set of simulations for 400 taxa where the only change is a four times as long sequence length of 1200.

In our time-stamped coalescent [27] simulations, we generated 100 phylogenetic trees and alignments for each of four different taxa sizes $n$: 40, 80, 160, and 320. Each tree coalescent process had a population size parameter ($\theta$) drawn from a log-normal distribution with a mean in log space of –2.4276, representing a mean in real space of approximately 0.09, and a standard deviation in log space of 0.5. The alignments consisted of 250 sites each. The youngest leaf was assigned age 0. The remaining leaf ages were distributed uniformly at random between 0 and 0.2. All other parameters were as in the Yule simulations.

We refer to the resulting datasets with `Coal40`, ..., `Coal320`, `Yule10`, ..., `Yule400`, `Yule400-long`. For each simulation, we ran 2 replicates with BEAST2 to obtain tree samples with 35k trees (50k trees for $n = 10$ and 20). In all cases, the replicates were checked to have run sufficiently long to ensure convergence, and excess burnin was discarded.

In addition, we perform an analysis on the `DS` datasets [28,29], specifically `DS1` to `DS4`; see Sect S1.3.2 of S1 Text for details.

## Results

We have presented a new tree distribution, CCD0, and introduced new point estimators. We now apply both CCD0 and CCD1 to the datasets described above to evaluate their point estimators and their performance as tractable tree distributions.

## Tree distributions

To evaluate the accuracy and precision of the CCDs and sample distributions, we used the datasets `Yule10` and `Yule20`. In Sect S1.3.2 of S1 Text, we also looked at `DS1` to `DS4` and included CCD2. For each simulation, we combined the 50k trees from the two replicates into one sample distribution of 100k trees, which acts as our (reference) *golden distribution*. These inference problems are relatively easy, and therefore, the probability of each tree (in particular, the high probability trees) is quite accurately estimated by the golden distributions.

However, for larger datasets with more taxa and higher complexity (in terms of entropy), achieving a "golden run" that accurately estimates tree probabilities within a reasonable time-frame is impossible. The size of tree space grows super-exponentially, and the probabilities of individual trees become exceedingly small, making it infeasible to estimate them based on their frequency in an MCMC sample. For instance, estimating the probability of a tree with a probability of $10^{-6}$ using an MCMC process that takes one second per sample would require at least $10^8$ samples, which would take over three years to complete. This estimate does not even account for the massive amounts of physical storage space needed to retain these tree samples, rendering it impossible to achieve such a golden run for problems more complex or notably larger than the Yule10 and Yule20 examples presented. In fact, this limitation of sample distributions is the very reason why we consider CCDs as estimates of the posterior tree distribution.

We used (sub)samples of size 3, 10, 30, 100, 300, 1k, 3k, 10k, and 30k to generate a CCD0, CCD1, CCD2, and sample distribution for each of the two replicates of all simulations – eight distributions per simulation. For each tree $T$ in the golden distribution, we then calculated the probability of $T$ in each of the eight distributions. Comparing these to the golden probabilities, we use different statistical measures to evaluate the accuracy of each distribution.

**Accuracy.**   For each sample size, we computed the mean absolute error (MAE) of tree probabilities for each distribution. In the context of a specific distribution (Sample, CCD0, CCD1, CCD2), the MAE is calculated as the average of the absolute differences in probabilities between this distribution and the reference (golden) distribution, considering all trees within the reference distribution. Note that the MAE weights the accuracy on high-probability trees more compared to lower probability trees. We then counted how often each distribution type had the lowest MAE, their number of *wins*. We further divided the simulations into five equal-sized groups (each of size 100) based on their entropy [12], that is, the sum of $-\Pr(T)\log\Pr(T)$ over all trees in the golden distribution. (For Yule10 the entropy bounds are set by 0.41, 1.76, 2.5, 3.25, 4.30, 7.68 with means of 1.20, 2.09, 2.84, 3.67, 5.30 and for Yule20 they are 0.09, 2.29, 3.22, 4.03, 5.08, 7.73 with means of 1.70, 2.82, 3.61, 4.52, 5.93.) Heatmaps of the wins in these categories for `Yule10` and `Yule20` are shown in Fig 4, where each tile is colored by the distribution that has the majority of wins and its win-% is given. A more detailed view on the number of wins can be found in Fig D in S1 Text.

We observe that there are three regimes based on the sample size: Roughly, from 3 to about 100 samples, CCD0 is the most accurate method; from 100 to 10k samples, CCD1 gives the best estimates; for the largest samples, CCD2 takes over CCD1. A lower entropy seems to prolong the dominance of CCD0. The boundaries of the regimes also vary with the problem size. The experiment confirms the regimes we expected: CCD0 is the simplest model and quickly provides a good estimate; CCD1 has more parameters, so needs longer to be saturated, whereas CCD0 then starts to show its bias. The same is true for CCD2, which needs even more samples than CCD1. In the long run, the sample distributions provide the best estimate, which can be observed by taking a more detailed look at the best performing distribution per simulation (cf. Sect S1.3.1 in S1 Text).

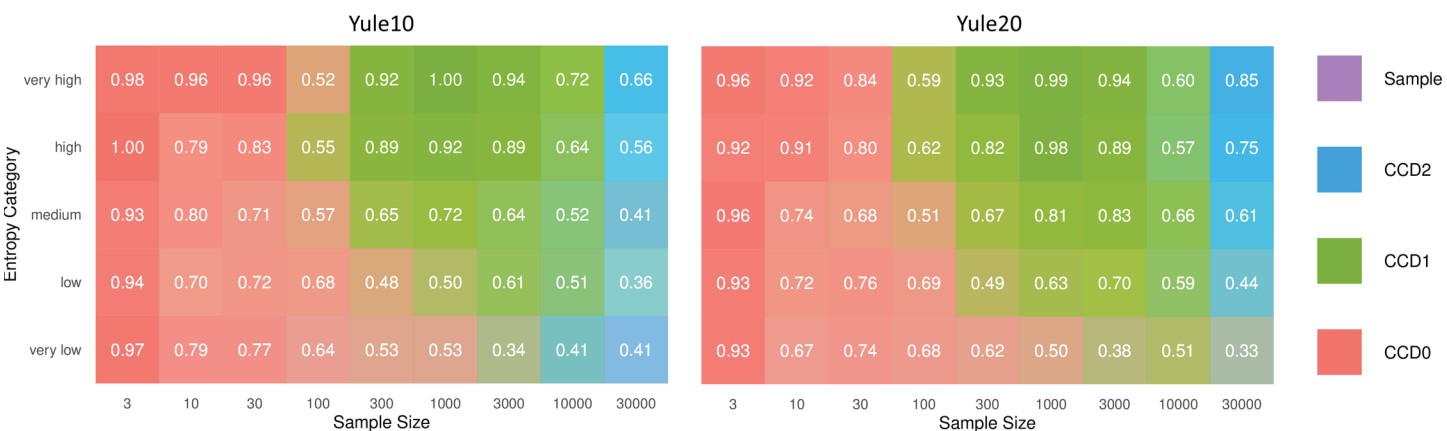

**Fig 4. Heatmap showing the majority wins based on MAE with simulations in five entropy categories (higher means noisier/harder); more saturated colors mean a larger wining margin for the respective distribution (CCD0, CCD1, CCD2 or the sample distribution).**

We also observe the regimes when we look at the mean relative error (MRE) of tree and clade probabilities; see Fig 5. (Since the results look very similar for `Yule10` and `Yule20`, those for `Yule10` can be found in Sect S1.3 of S1 Text.) The MRE is defined as the mean of the absolute difference in probability between the golden distribution and the generated distribution (CCD0, CCD1, CCD2, or Sample) divided by the golden probability for all trees/clades. Note that for the MRE, a small absolute difference in probability for low probability trees causes a larger relative error. Since tree probabilities in the tail of the distribution are not that well estimated, we consider thus only the trees in the 50% and 95% credibility intervals. That is, the minimum number of highest ranked trees in the golden distribution whose probabilities sum up to 50%/95%. For clades we consider all clades in the golden distribution. For small sample sizes, CCD0 performs better/equal than CCD1 up to about sample sizes of 30/300. Note that CCD0 then does not improve any further, indicating the limitations of the CCD0 model. The performance of CCD1 remains the best even for larger sample sizes with CCD2 close behind and the sample distribution only slowly catching up. Note that in the case of clade probabilities, we have merged CCD1 and Sample because they are the same.

Looking at the mean estimated rank of the top tree of the golden distribution in the other distributions for each simulation reveals a similar picture; see Fig 6. CCD0 is best for sample sizes up to and including 30, but above 100 CCD1 and CCD2 perform better on average; the sample distribution requires 1k samples to become competitive.

**Precision.** To evaluate the precision, we computed the difference in the tree probabilities between the two replicates for each sample size. The mean over the 100 simulations for `Yule20` are shown in Fig 7. We observe that the CCDs consistently show a higher precision than the sample distribution for all sample sizes. Note that high precision also implies a high stability.

**Representativeness.** Note that by construction for a given MCMC run, if the sample distribution contains the true tree then so do the CCDs; analogously, if CCD1 (CCD2) contains the true tree then so does CCD0 (resp. CCD1 and CCD0). Table 1 shows how the percentage of the distributions (both replicates per simulation) that contain the true tree for the 250/100 simulations of the `Yule20` and `Yule50` dataset. For the former, we observe that CCD0 and CCD1 cross the 95% threshold already for 100 samples, while the sample distribution only does so at 3k samples. The difference becomes even more apparent for `Yule50`, where

the sample distribution only reaches 3.5% with 30k sampled trees, while the CCDs quickly contain the true tree in the majority of simulations and also reach the 95% threshold.

## Point estimators

We evaluated the point estimators based on the following properties. Firstly, we have the *accuracy* – a good point estimate should be close to the truth (low Root Mean Squared Error or average distance; testable in simulation studies). Further, we can measure the behaviour under

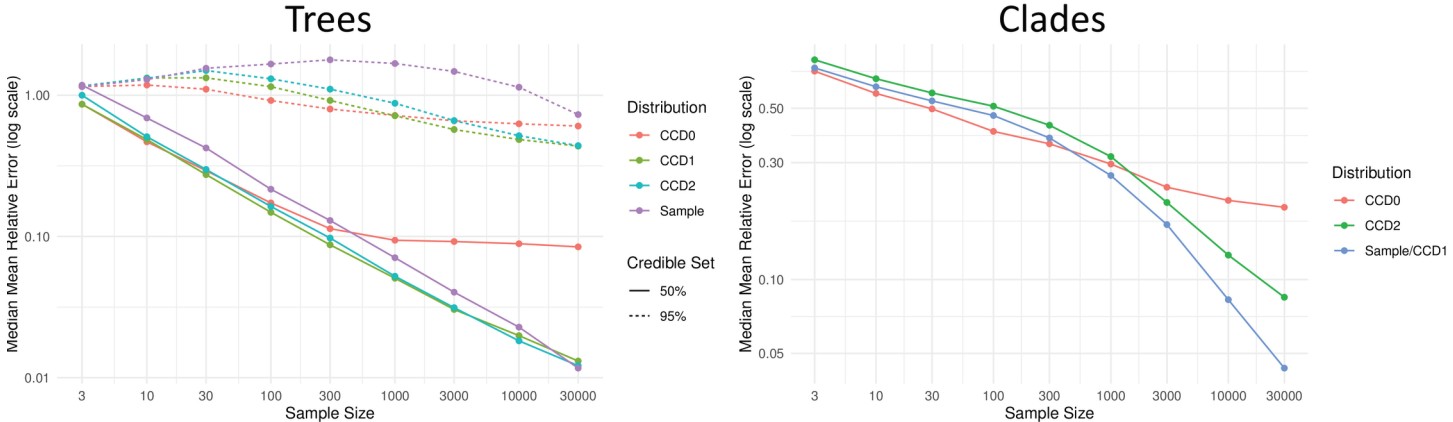

**Fig 5. Median MRE for trees and clades in the golden distribution per sample size for `Yule20`.** Trees are separated into 50% and 95% credible sets.

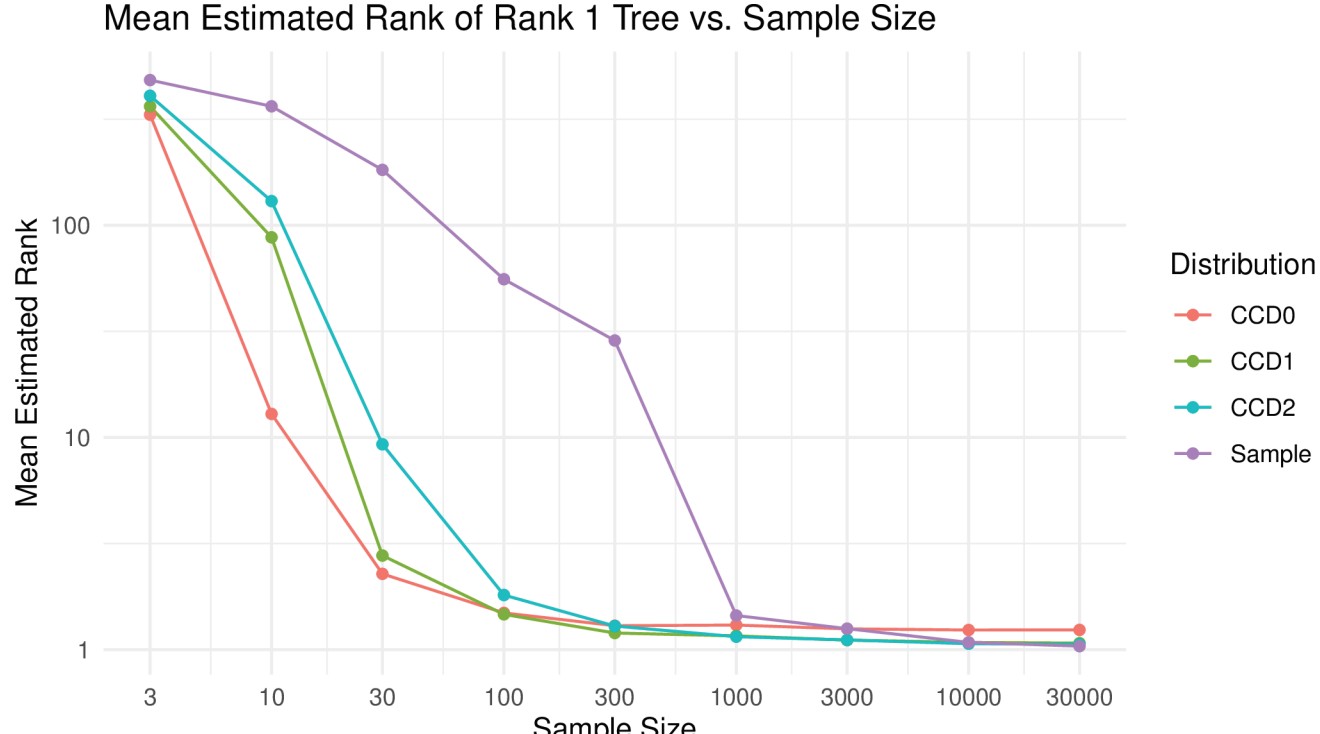

**Fig 6. Mean rank of the top tree (rank 1) in the golden distribution in the other distributions per sample size for `Yule20`.**

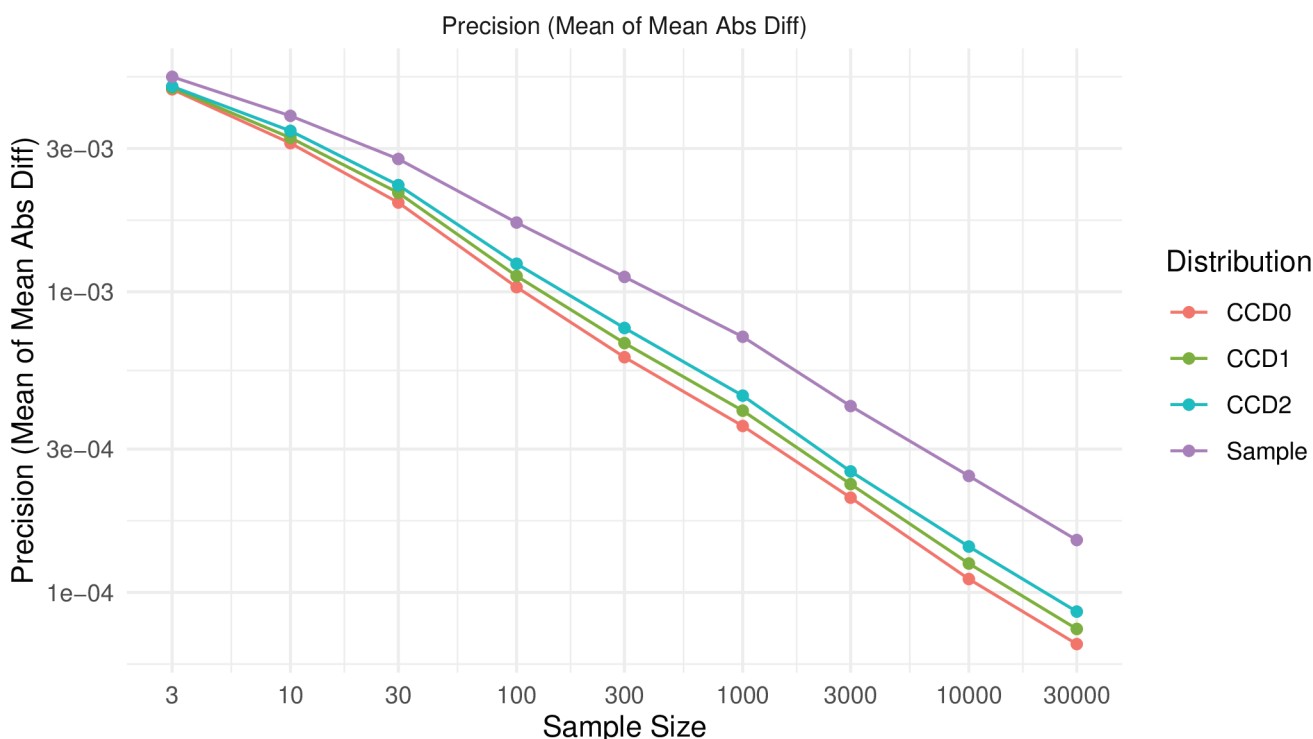

**Fig 7. Evaluating the precision of the distributions, we computed the mean of mean absolute differences of tree probabilities by the distributions between two replicates per sample size for `Yule20`.**

**Table 1. Percentage of the true tree being contained in a distribution for `Yule20` and `Yule50` (out of the 250/100 simulations with 2 replicates each).**

|  | | Sample Size | | | | | | | | |
|---|---|---|---|---|---|---|---|---|---|---|
|  | Distribution | 3 | 10 | 30 | 100 | 300 | 1k | 3k | 10k | 30k |
| **Yule20** | CCD0 | 39.8 | 75.8 | 90.8 | 96.8 | 99.2 | 99.8 | 100 | 100 | 100 |
|  | CCD1 | 37 | 67.8 | 86.8 | 95.6 | 98.4 | 99.6 | 100 | 100 | 100 |
|  | CCD2 | 33.2 | 60 | 83.2 | 93.6 | 97.2 | 98.4 | 99.6 | 99.8 | 99.8 |
|  | Sample | 23.2 | 41 | 60.6 | 76.6 | 87.6 | 92.8 | 95.6 | 98.2 | 98.8 |
| **Yule50** | CCD0 | 0 | 9.5 | 46 | 70.5 | 87.5 | 94.5 | 100 | 99.5 | 100 |
|  | CCD1 | 0 | 3.5 | 24.5 | 50.5 | 69.5 | 80.5 | 90 | 98 | 99.5 |
|  | CCD2 | 0 | 1 | 11.5 | 32 | 53.5 | 69.5 | 79.5 | 91.5 | 95 |
|  | Sample | 0 | 0 | 0 | 0 | 0 | 0 | 1.5 | 2.5 | 3.5 |

different MCMC replicates; a good point estimator should be *precise* (small distance between estimates) and, related to that, *stable* (consistent distance to truth).

Holder et al. [30] argued for MRC trees as point estimates by showing that if we define a loss function with penalties for missed and wrong clades, then the MRC tree tries to minimize the loss. In fact, if we only report fully-resolved trees, then this is equivalent to the well-known Robinson-Foulds (RF) distance [31]. Recall that the *Robinson-Foulds (RF) distance* of two trees $T$ and $T'$ equals the symmetric distance of their clade sets $\mathcal{C}(T)$ and $\mathcal{C}(T')$, which we here always divide by two. So for fully-resolved trees, the RF distance to the truth measures how many clades the point estimate gets wrong. As we discuss later, we get few non-fully-resolved trees from the greedy consensus method and thus do not compensate for the fact that these benefit from having less clades that can contribute to the symmetric difference;

For our experiments, we used the datasets `Yule50` to `Yule400` and `Coal40` to `Coal320`. For each simulation and each of the two replicates, we again used samples of size 3, 10, 30, 100, 300, 1k, 3k, 10k, and 30k to generate a CCD0, a CCD1, and a sample distribution each. With the CCDs we computed the CCD-MAP trees and the CCD0-MSCC tree, and based on the sample distribution we computed the MCC tree and the greedy consensus tree. As reference we have the *true tree*, the one used to generate the alignments, of each simulation. (We only show the results for the for larger datasets here; those for the four smaller datasets are very similar and thus only given in Sect S1.4 of S1 Text.) In addition, we also considered the tree topology of the state with highest (sampled) posterior density as a topological point estimate (the posterior density of the state includes consideration of the divergence ages and other model parameters). However, since it performed consistently worse than even the MCC tree and since the CCD0-MSCC tree behaved almost exactly as the CCD0-MAP tree we excluded them from the figures to improve visual clarity.

**Accuracy.** Fig 8 shows the mean relative RF distance of the point estimates to the true tree for different sample sizes. The relative RF distance describes the percentage of the $n-2$ clades of the true tree an estimator got wrong. So for e.g. `Yule400`, a relative RF distance of 0.16 (0.1) means that about 64 (resp. 40) of 398 nontrivial clades are different from the true tree. We observe that CCD0-MAP performs best from 3 to 30k trees. At around 30 to 100 trees for the Yule simulations and around 100 to 300 trees for the coalescent simulations greedy consensus catches up and performs equally well. CCD1-MAP gets close to this performance but does not fully catch up. MCC on the other remains at least 1% behind the top estimators.

**Precision.** To evaluate the precision, we computed the mean distance between the point estimates of two corresponding replicates; see Fig 9. We observe that greedy consensus and the CCD based methods have significant higher precision than MCC, with CCD1-MAP lacking slightly behind the others. For 1k trees, the CCD0 estimators and greedy consensus vary in less than 10 clades between replicates, whereas MCC varies five- to tenfold of that. Note that a high precision also implies a high stability (variance in distance to the true tree).

**Running time.** We also want to report on the running times of our implementations for the largest dataset `Yule400`. Constructing a CCD1 and CCD0 on samples of 30k trees (which requires parsing the file with 35k trees) took on average 90 seconds, the same as constructing the MCC tree. Computing any of the other point estimates took only few milliseconds. The bottleneck seems to be parsing the large file and not the construction.

**Resolvedness.** We also tested in how many simulations the greedy consensus tree was not fully resolved. The results (see Table A in S1 Text) show that for 300 trees and more, the greedy consensus tree was always fully resolved on our datasets; for 30 trees, the greedy consensus tree was not fully resolved at most 4.5% of the simulations. Note that with better Monte Carlo estimates of clade frequencies, ties that can cause unresolved trees become less likely.

## Discussion

The CCD approach can be described as a bias-variance trade-off in the context of MCMC summarization. These tractable tree distributions exhibit a certain level of bias (due to the independence assumptions employed) in exchange for reduced variance in the estimates when faced with Monte Carlo error, particularly in cases of low ESS (relative to the posterior variance in tree space). The number of parameters of the models grow from CCD0 (clades) to CCD1 (clade splits) to CCD2 (pairs of clade splits) and finally to sample distributions (trees), demanding an increasing number of trees to estimate them accurately. This is confirmed by our experiments on easy and small simulated problems, where we observed these

## Mean distance to truth vs. Sample Size for Different Estimators

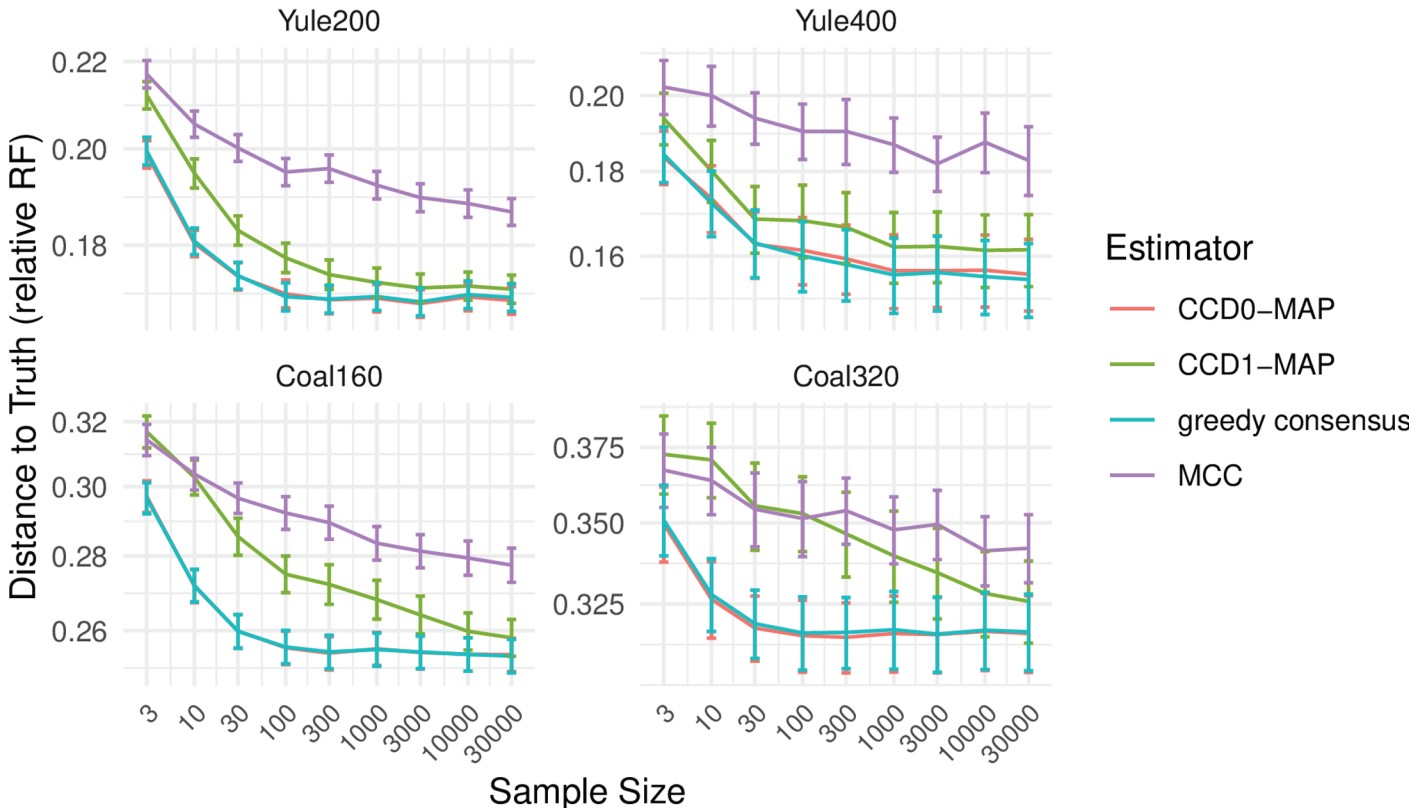

**Fig 8. The accuracy of the point estimates measured in terms of the mean relative RF distance to the true tree for different sample sizes of the large datasets.**

regimes in terms of the number of sampled trees: First, CCD0 is best for few samples in terms of accuracy, precision, and stability, then CCD1 catches up and becomes the best method in the mid range, while CCD2 and the sample distribution require a huge number of sampled trees to become competitive. Unsurprisingly, the bias of CCD0 becomes apparent with a large enough number of uncorrelated samples. On the real datasets `DS1` and `DS4` (which have low entropy), the richer models of CCD1 and CCD2 can capture the structure of the posterior better than a CCD0, even with a small number of sampled trees. For all `DS1` to `DS4`, the sample distribution is even competitive or the best choice for large number of trees. However, for non-trivial problems, sampling enough trees with MCMC to reach the regimes of CCD1, CCD2, or sample distributions may not be feasible. In such cases, advanced methods such as regularization used by Zhang and Matsen [17] promise to be beneficial; they found that their sDAG model performs better than CCD1 in capturing posterior distributions of real datasets. While our experiments suggest that CCD0 offers the overall best posterior estimate for hard simulated problems (in terms of entropy), the question remains of how to select the best model for real datasets. This is particularly relevant when datasets are relatively simple (in terms of entropy) or exhibit challenging characteristics such as multi-modality [32]. One way to assess this could be the use of the Akaike information criterion (AIC), the Bayesian information criterion (BIC), or cross-validation. In our preliminary experiments (see Sect S1.3.2 in S1 Text), the AIC scores align well with the accuracy of the distributions. Note though AIC

## Precision vs. Sample Size for Different Estimators (RF)

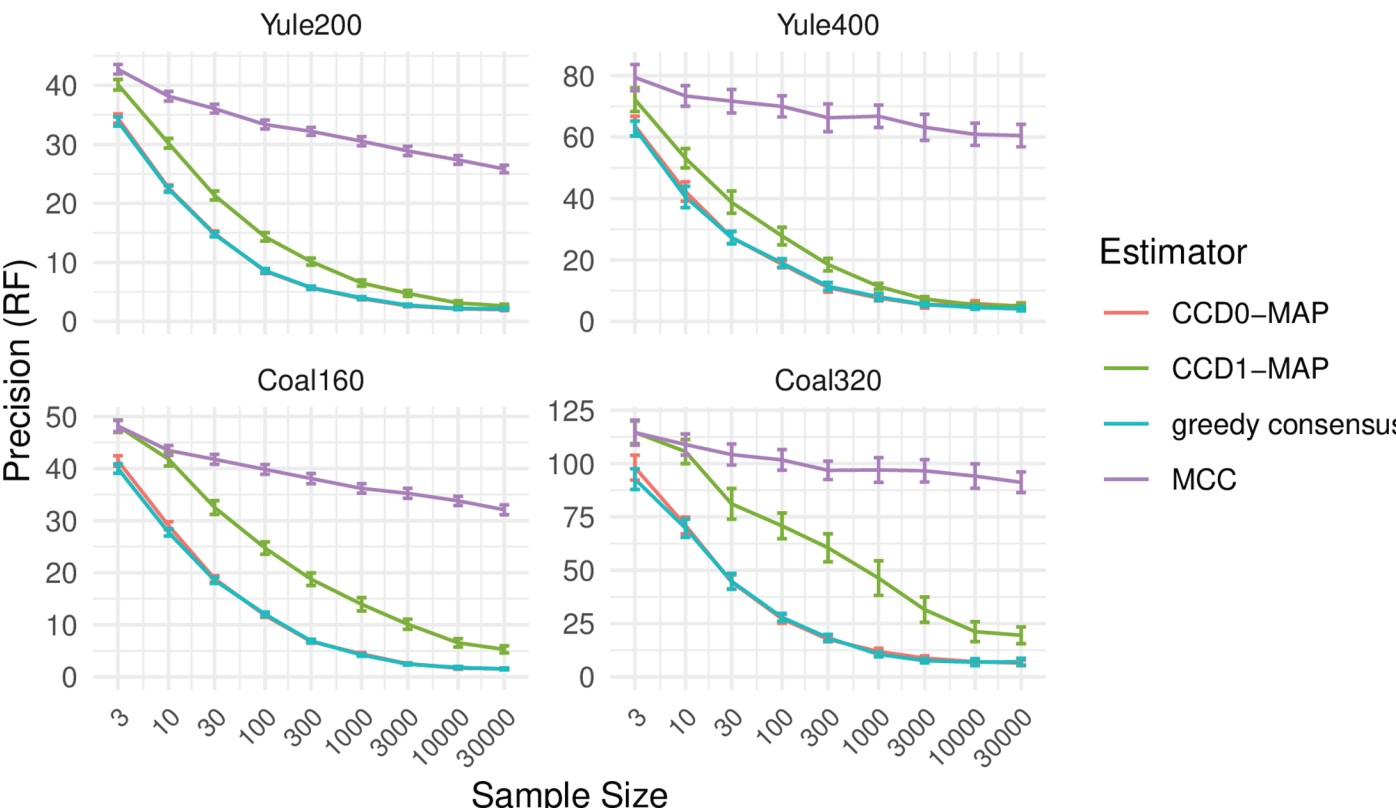

**Fig 9. The precision of the point estimates in terms of the RF distance, that is, the mean RF distance of the point estimates of the two replicates of each simulation.**

requires independent samples to be meaningful, which in the context of MCMC necessitates good estimates of the effective sample size (ESS). However, computing the ESS of trees remains an ongoing challenge [33].

With an implementation that uses CCD graphs, many tasks related to tree distributions can be performed efficiently (fixed-parameter tractable in the number of clades and clade splits). This includes sampling a tree, computing the probability of a tree, as required for example for BCA [13,15], computing the MAP tree, and computing the entropy and the number of trees in the distribution. In practice, the running time is dominated by parsing the trees while building the CCD, whereas computing the MAP tree takes negligible time. While for large and very diffuse (more prior-like) distributions the construction of a CCD0 may take noticeable time (minutes), it would still only be a fraction of the days or weeks needed to compute such a distribution via MCMC.

Concerning the point estimates, we demonstrated that the CCD-MAP trees and the greedy consensus tree outperform the commonly used MCC tree in terms of accuracy and precision. So not only do they produce better trees in general, but they are also more robust to the random sampling process of MCMC. This finding is concerning given that the MCC tree has been the standard point estimate used by almost every BEAST practitioner for decades. Additionally, we find that the CCD0-MAP tree performs equally or better than the greedy

consensus tree, with the added benefit that both variants of the CCD-MAP tree guarantee a fully resolved tree. While getting an unresolved greedy consensus tree may not be an issue for many problems (cf. Table A in S1 Text), we want to point out that (i) in viral phylodynamics, it is typical to encounter (near)identical sequences resulting in partially diffuse posteriors, thus increasing the probability of encountering unresolved greedy consensus trees, and (ii) finding the most resolved greedy consensus tree is an NP-hard problem. The CCD1-MAP tree does not match the accuracy of the CCD0-MAP tree in our experiments on non-trivial problems, since even for large samples we do not reach the CCD1-regime observed in smaller analyses. On the Yule20 dataset, we could not observe a performance difference between the CCD1-MAP tree and CCD0-MAP tree. For a sufficiently large number of uncorrelated samples, the CCD1-MAP tree is expected to perform equal or even better than the CCD0-MAP tree.

Suppose that we had the true posterior distribution. This distribution is not only on tree topologies but also on branch lengths and other model parameters. Then we would take the actual MAP tree, that is, the tree with maximum posterior probability after marginalising (i.e., integrating) over the branch lengths and the other model parameters. Estimating the posterior distribution with an MCMC sample, the marginalising happens automatically if we just look at the frequencies of different tree topologies. However, if the probability of the most probable tree is less than $1/k$ (for sample size of $k$), the Monte Carlo sample counts will not accurately reflect the posterior probabilities of individual tree topologies, since every tree topology is sampled either 0 or 1 times. Due to this limitation, we require other summary methods (as described in this paper). In the past, the tree topology from the sampled state with highest posterior density has been extracted (without marginalising over other parameters) [34]. However, our experiments showed that this point estimate of tree topology performs even worse than the MCC tree, another commonly used summary method. Our recommendation is thus to use CCD0-MAP tree and the greedy consensus tree.

Despite the existence of various tree metrics our evaluation focuses on the Robinson-Foulds distance. This choice is justified because all the point estimates compared in the paper – CCD-MAP, MCC, and greedy consensus – are primarily based on constructing a topology. Hence, the Robinson-Foulds distance is particularly suitable for evaluating their performance, especially in the context of systematics, where one of the primary goals of a phylogeny is to obtain accurate clade information [30]. In this context, the Robinson-Foulds metric directly quantifies the performance when comparing a point estimate to the true tree.

## Conclusion

This research has shown that the CCD0-MAP tree and the greedy consensus should be the preferred point estimators for Bayesian phylogenetic inference of time-trees. The restriction to sampled trees comes at such a high cost that previous caution against using unsampled trees as point estimates is not warranted. We can thus retire the MCC-from-sample point estimator. Furthermore, CCDs offer better estimates of individual tree probabilities than the sample distribution for hard problems. However, picking the right CCD model for a particular dataset remains a tricky problem that requires further research.

Our conclusions are primarily drawn from well-calibrated simulation studies, which are the current standard for evaluating phylogenetic tree tools. However, the extent to which these simulations accurately represent real datasets and their posterior distributions remains uncertain. We have designed our simulations to capture a broad range of entropies observed in real datasets, aiming for greater realism. Nevertheless, these simulations do not account for more complex scenarios, such as multi-modality or non-standard distributional shapes.

Addressing these challenges will require significant advances in the field, as the complexities of phylogenetic treespaces are not yet fully understood and demand further research.

While our approach was developed mainly for TreeAnnotator within the BEAST2 framework [1], our results are applicable to any sample of rooted tree topologies that represents a posterior distribution. Furthermore, it would be straightforward to incorporate support for unresolved trees. We have incorporated CCD-based point estimators into the existing TreeAnnotator software, providing the CCD package, which enables users to easily access and use this new method on their data.

In practice, time information of point estimates is also of great interest. The CCD-based point estimates fit in the commonly used framework of estimating the tree topology first followed by annotating it with divergence ages. These latter methods are independent from CCDs. It would be interesting to see how greedy consensus and the CCD0-MAP tree combined with an annotation method perform in comparison to other combined approaches and to methods that estimate the topology and branch lengths at the same time, like the matrix method [35].

We hope to use and further develop CCDs for other tasks when working with posterior distributions. This includes the computation of the credibility set of tree topologies, MCMC convergence analysis (cf. Berling et al. [29]), and detection of rogue taxa. One further interesting avenue is to investigate how the parameters of a CCD could be populated efficiently with other means than observed sample frequencies, for example with maximum likelihood or variational methods [18,19].

## Supporting information

**S1 Text. Supporting text with embedded additional materials.** Containing 4 supporting sections describing further details and results, 15 additional figures, and a table. (PDF)

## Acknowledgments

We would also like to thank Jordan Douglas for his help setting up simulation studies as well as Erick Matsen for his helpful comments and suggestions.

## Author contributions

**Conceptualization:** Lars Berling, Jonathan Klawitter, Alex Gavryushkin, Alexei J. Drummond.

**Data curation:** Lars Berling, Jonathan Klawitter, Remco Bouckaert, Dong Xie.

**Formal analysis:** Lars Berling, Jonathan Klawitter.

**Funding acquisition:** Alexei J. Drummond.

**Investigation:** Lars Berling, Jonathan Klawitter.

**Methodology:** Lars Berling, Jonathan Klawitter, Alexei J. Drummond.

**Project administration:** Lars Berling, Jonathan Klawitter, Alexei J. Drummond.

**Resources:** Lars Berling, Jonathan Klawitter.

**Software:** Lars Berling, Jonathan Klawitter, Remco Bouckaert, Alexei J. Drummond.

**Supervision:** Alex Gavryushkin, Alexei J. Drummond.

**Validation:** Lars Berling, Jonathan Klawitter.

**Visualization:** Lars Berling, Jonathan Klawitter, Alexei J. Drummond.

**Writing – original draft:** Lars Berling, Jonathan Klawitter.

**Writing – review & editing:** Lars Berling, Jonathan Klawitter, Remco Bouckaert, Alex Gavryushkin, Alexei J. Drummond.

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
