## [Decision Letter · Decision Letter 0]

16 Apr 2024

Dear Mr. Berling,

Thank you very much for submitting your manuscript "Accurate Bayesian phylogenetic point estimation using a tree distribution parameterized by clade probabilities" for consideration at PLOS Computational Biology.

As with all papers reviewed by the journal, your manuscript was reviewed by members of the editorial board and by several independent reviewers. In light of the reviews (below this email), we would like to invite the resubmission of a significantly-revised version that takes into account the reviewers' comments.

We cannot make any decision about publication until we have seen the revised manuscript and your response to the reviewers' comments. Your revised manuscript is also likely to be sent to reviewers for further evaluation.

Sincerely,

Jennifer A. Flegg

Academic Editor

PLOS Computational Biology

Natalia Komarova

Section Editor

PLOS Computational Biology

Reviewer's Responses to Questions

**Comments to the Authors:**

Reviewer #1: * Summary

** General summary

This manuscript describes a novel tree distribution, CCD0 (a variation

on the existing CCD1 distribution), and extensively tests its

performance on some standard tasks. The use case for CCD0/1 includes

summarising a set of tree topologies (for example an MCMC sample) into

a single object that is computationally easy to work with.

Importantly, CCD0/1 give us a way to generate a point estimate of a

tree topology. There are existing ways to do this, MRC and the

(dominant) MCC, but it is demonstrated via extensive simulation

studies that the CCD0/1 methods produce better point estimates (under

appropriate metrics).

This work culminates in the claim that we "can thus retire the

MCC-from-sample point estimator." While I agree with the spirit of

this statement, I suspect MCC, as the incumbent will persist for a

while. However, having CCD-MAP in TreeAnnotator will certainly help

with the transition.

** Technical summary

Here is my understanding of the technical details. My comments are

based on this.

The CCD0/1 are constructed by taking a set of tree topologies and

building a forest network. The forest network is then converted to a

CCD graph (which defines a distribution over a set of topologies). In

the case of the CCD1 the probability masses come from the empirical

distribution of clade splits. In the case of CCD0 the probability

masses come from the empirical distribution of clades (after a complex

set of calculations which I will discuss below). The result is that

the novel CCD0 is a more stable (though also more biased) construction

of a CCD graph than the existing CCD1 distribution.

The CCD graph enables efficient computation for a range of standard

tasks. I am surprised (and slightly concerned) that the CCD1 has not

gained more traction in the years since its publication, but the

tooling developed and released here will make this methodology usable

by the large audience of BEAST practitioners. Hopefully CCD0 will gain

more adoption. There is *extensive* simulation study to demonstrate

the efficacy of the method in comparison to currently used

alternatives.

* Recommendation

The CCD0 described represents a substantial improvement in how we

handle posterior samples in Bayesian phylogenetics. When it becomes

adopted (which I am optimistic will happen soon) it will lead to an

improvement in the generation of topology point estimates, (and

despite being a technical manuscript will generate a large number of

citations for PCB). I like this manuscript, the construction of the

CCD0/1 are a very good idea and will improve Bayesian phylogenetics. I

want this published. However, I have several major comments I would

like addressed before I would consider this ready for publication, in

particular #1 and #2. Addressing them will require substantial

revisions.

* Comments

** Major comments

1. The Remark, lines 190--195, is very concerning. If this is only an

issue for CCD1, then that should be spelled out very clearly. If it

is an issue for both CCD0 and CCD1 (which I think is the case) then

this remark should be moved to a position where it is clearer that

it applies to both and pointed out again in the Discussion. If you

have a fool-proof remedy for this, I would really like to see that

spelled out in detail. If you do not have a good remedy, then

please remind the reader of this in the Discussion as it is

something that must be kept in mind.

2. Lines 203--227 are obviously important, and I think I understood

them after a lot of effort (and frustration). I think the paper

will be much improved if you a.) replace this section with a high

level overview of the motivation for CCD0 and how it differs from

CCD1; and b.) put this section into an appendix, and elaborate and

simplify the derivation for struggling readers (such as myself).

This way a reader can get a feel for the method (without being

scared off) and readers interested in details can enjoy an expanded

derivation in the appendix.

3. Figure 2 is very helpful, but I think you can help the reader by

leaning on it a little more. For example, there is a lot of

notation, you could give examples of a lot more of the notation by

spelling it out with this example. Also, I think it will be easier

to see the bipartite structure if you use more distinct colours in

Figure 2.

4. Optional but important: To hasten uptake of CCD0 over MCC, please

consider putting together a Taming the BEAST tutorial with a clean

example.

** Minor comments

1. Figure 5 is great!

2. Line 489: This is a really impactful statement, it could probably

be repeated more explicitly in the introduction.

3. Line 517: "caution against using" not "caution of using", I'm not

sure.

4. Line 49: I think it would be helpful to introduce the acronym "MRC"

here.

5. Line 173: I find the vertical bar to indicate a split to be

distracting since it is close to conditioning when used within a

probability, perhaps you could consider using hyphens instead.

6. Line 285: Notation is unpleasant in Equation (1) here please give

extra consideration to hyphen or similar.

7. Line 347: Six distributions? I got a bit muddled on this line, why

six?

8. I see there is a blog post on getting this method running

efficiently, please consider putting a copy of this as an appendix

if there is significant scholarly content there.

9. Line 188--189: Possible inconsistency in the notation?

10. I think it should be reasonably straightforward to represent

multifurcating trees, it might be worth pointing this out in the

discussion, but obviously not necessary.

11. Line 199: this sentence is a bit awkward and will be cleaner if you

reorder some of the words.

12. Line 204: Consider pointing out that this is the power set.

13. Line 90: I'm disagree that other trees should *not* be represented

here. I would have thought it was useful to have the trees

assigned a very low probability so that I know it is accounted for

by the method, i.e. it is known to be low probability, not an

oversight. Perhaps you expand on this to explain your stance, am I

missing something?

14. Line 257: Maybe include the base case for the recursive calculation

of entropy too?

15. Line 83: It might help to indicate if you intend here to give a

formal definition of "tractable tree distribution" here, or if this

is just a short hand for a useful distribution.

16. Line 45: "stickiness" seems a bit vague.

17. References: Some of the preprint server references are missing DOIs

18. References: Some of the journal titles have incorrect capitalisation

19. References: Some of the article titles have incorrect capitalisation

Reviewer #2: The authors have provided a new parameterization of distributions on tree probabilities (CCD0) as well as interesting simulations concerning the performance of the original conditional clade distribution framework. They also implement MAP point estimates for these distributions. They find that these outperform the very commonly used maximum clade credibility tree.

The paper is interesting, well structured, executed, and written.

My number one issue with this paper is that it makes quite a sweeping recommendation using only simulated data, and easy simulated data for that matter. It's important to note that for the most part, the evaluation of these models doesn't have to happen on simulated data, and instead can happen on real data with all of the quirks of real data. Because the simulation framework is very simple and so it's not perhaps not so surprising that the simplest estimator, CCD0, performs well. However, I wouldn't feel very confident in extending these conclusions to real data. In our work, using real data with multimodal posteriors, we have found that an even more complex estimator than CCD1 is needed to accurately capture tree posterior distributions.

Speaking of which, we have developed a more general framework than CCD1, including regularization and unrooted trees. See

Zhang, C., & Matsen, F. A., IV. (2018). Generalizing Tree Probability Estimation via Bayesian Networks. In S. Bengio, H. Wallach, H. Larochelle, K. Grauman, N. Cesa-Bianchi, & R. Garnett (Eds.), Advances in Neural Information Processing Systems 31 (pp. 1449–1458). Curran Associates, Inc. http://papers.nips.cc/paper/7418-generalizing-tree-probability-estimation-via-bayesian-networks.pdf

This work uses the terminology "subsplit" for "clade split".

Perhaps the most relevant part of this prior work is that we can think of these distributions as parameterizations of probability distributions on trees, and that we can use statistical methods to estimate parameters of these models. Specifically, one can use regularization to improve model fit. That idea doesn't seem to have made it into the current paper.

We have also described how these models can be expressed as a graph:

Jun, S.-H., Nasif, H., Jennings-Shaffer, C., Rich, D. H., Kooperberg, A., Fourment, M., Zhang, C., Suchard, M. A., & Matsen, F. A., 4th. (2023). A topology-marginal composite likelihood via a generalized phylogenetic pruning algorithm. Algorithms for Molecular Biology: AMB, 18(1), 10. https://doi.org/10.1186/s13015-023-00235-1

CCD0 is an interesting advance, specifically how it incorporates clade splits that don't exist in the original data set. However, it breaks some of the conditioning that can be useful. I wonder if there is something between CCD1 and CCD0, in which one could use the CCD1 conditional probability when that's possible, but uses the CCD0 probability when not. Or some balance between the two.

Details:

The introduction is written in a BEAST-centric way.

* I suggest that it would be worth noting that consensus trees have other applications than Bayesian point estimates

* L39: unrooted phylogenetic inferences don't have divergence times

It would be nice to refer to combine figure 1 and figure 2 into a single figure and refer to them right away when introducing the forest network, rather than waiting until introducing the example. It'd be great to expand the figure legend to describe the symbology for \mathcal{C} and \mathcal{S}.

"Does removing a tree T from the CCD1 and thus decreasing clade and clade split frequencies by one, significantly change the probability of T ." Can't we just look for edges of probability 1/k?

L203: It would be helpful to clarify the goal of this paragraph more precisely than "based on a function F".

MRE: please specify what this means exactly

Figure 5: please identifiy the labels S, CCD0, CCD1 in the figure legend.

Why use a multiple of 100 for the yule simulations, but not for the coal simulations?

Reviewer #3: Dear Dr Berling,

I have read your manuscript on "Accurate Bayesian phylogenetic point estimation using a tree distribution parameterized by clade probabilities". My overall impression is that this is a nice manuscript that will be useful to the community. I really liked it and while reading the manuscript I had several questions and ideas. So please see my comments below as suggestion from an interested reader instead of a critical reviewer.

My main comment regarding the manuscript is the omission of the MAP tree as a point estimator. Personally, I would always choose the MAP tree over the MCC or consensus tree, especially if those are restricted to correspond to a sampled tree. If I believe that my MCMC samples are representative of the posterior distribution, then why not choose the tree with the highest posterior probability? My suggestion is therefore to include the MAP tree in the evaluations (Fig 5).

If I understood the results correctly, then the CCD0 and CCD1 provide better estimates of the posterior probability of the MAP tree (or true tree?) than the simple sample frequency from the Monte Carlo samples. If this is true, then I would conclude that it is always better to use the CCD0 or CCD1 to estimate the posterior probabilities of the MAP tree. Perhaps this could be emphasized?

Perhaps you could add a small subsection about the evaluation of your simulations in the methods section, especially including an equation for the MAE. This is even more important for the MRE. How did you use the credible interval for these plots (Fig 6)?

Would it be possible to plot the actual MAE and not only the wins? You could keep the same x-axis as in Fig 5, and plot 3 lines (or 3-times 5 lines) with the MAE for each method. The winning estimator could still be pretty bad. Maybe this would be similar to Fig 7?

From Table 1 it seems that the posterior distribution is relatively flat. Would it be possible to perform an experiment with an increasing number of sites in the alignment for the Yule50 trees to see the impact of the number of trees in the CI on the recovery frequency of the true tree? Similarly, do you think you could plot the posterior probability of the true tree vs frequency of the true tree being contained?

Would it be possible to include the MAP tree as an estimator in Figure 9? Or can you say how it would perform?

How specifically did you treat unresolved nodes in the computation of the RF distance? I'm specifically wondering because some implementations would give the star tree a 0-distance, which means that any consensus method that is ultra conservative will be less penalized.

I know this might be asking a bit much, but do you think it would be possible to evaluate the estimators on empirical datasets? For example, you could use the very long runs to obtain a "true" tree and then compare the RF distance between the different tree estimators.

Could you add some recommendation about how many trees one would need to sample? Hopefully no one would sample fewer than 100 trees.

Since you mentioned in line 294 that CCD's could be used to obtain posterior probabilities of clades (or split frequencies), would it be possible to evaluate how well the CCD's perform compared with standard sample frequencies? Given that many people use posterior probabilities of clades to assess support, it would be nice if CCD's are more robust than simple Monte Carlo frequencies.

Finally, I was wondering if the CCD0 and CCD1 would perform better than sample frequencies from independent MCMC samples. Maybe I missed this in the discussion/conclusion?

Minor comments:

- line 43: Please define BHV

- line 282: dynamic program?

- line 286: Is the CCD-MAP the same as the MCC except using conditional clade probabilities? Maybe that could be explained in the text.

- line 326: 2 chains -> 2 replicates (chains could also mean one cold and one heated chain, so better use replicates to avoid confusion).

- line 327: chains -> replicates

- Fig 5: Please specify S in the caption

- Fig 7: What is the golden mode tree?

**Have the authors made all data and (if applicable) computational code underlying the findings in their manuscript fully available?**

Reviewer #1: Yes

Reviewer #2: Yes

Reviewer #3: Yes

PLOS authors have the option to publish the peer review history of their article (what does this mean?). If published, this will include your full peer review and any attached files.

Reviewer #1: No

Reviewer #2: **Yes: **Frederick "Erick" Matsen

Reviewer #3: No
---

## [Decision Letter · Decision Letter 1]

19 Aug 2024

Dear Mr. Berling,

Thank you very much for submitting your manuscript "Accurate Bayesian phylogenetic point estimation using a tree distribution parameterized by clade probabilities" for consideration at PLOS Computational Biology.

As with all papers reviewed by the journal, your manuscript was reviewed by members of the editorial board and by several independent reviewers. In light of the reviews (below this email), we would like to invite the resubmission of a significantly-revised version that takes into account the reviewers' comments.

** As you will see, one of the reviewers still has some substantial reservations about the manuscript. Please address all the comments of both referees.

We cannot make any decision about publication until we have seen the revised manuscript and your response to the reviewers' comments. Your revised manuscript is also likely to be sent to reviewers for further evaluation.

Sincerely,

Natalia L. Komarova

Section Editor

PLOS Computational Biology

Natalia Komarova

Section Editor

PLOS Computational Biology

Reviewer's Responses to Questions

**Comments to the Authors:**

Reviewer #1: Given there has been a substantial extension to the work, through the application to real datasets, I would like to make a couple more comments:

1. I would like the application to real data to appear in more detail in the results section. This is important for conveying the limitations of the CCDs. My read on this is: for complex problems the sample based answer is preferable *if* you can get a sufficient number of posterior samples, *but* the CCDs work better if you are limited to a small sample. If this is the correct interpretation of your results, I think it needs to be spelled out very clearly in the Discussion and the Conclusion.

2. The application to real data isn't mentioned in the abstract or introduction, which seems odd. Please include a brief mention of these case studies in both the abstract and the introduction. It is a nice addition, you should point it out to people!

Otherwise, the authors have satisfactorily addressed all my previous comments. Congratulations on a great manuscript!

Reviewer #2: Major points:

1.

One of the questions addressed by the manuscript is which model (here denoted CCDX) is most appropriate for phylogenetic posterior distributions. In my previous review, I suggested that the manuscript would be improved using an analysis using real data. The revised paper now has a limited analysis in an appendix. Out of the 8 DS data sets, the authors have done the first four, two of which (DS2 and DS3) have quite small posterior distributions (in Table 1 of Zhang and Matsen 2018, these are the only data sets that have < 100 trees in a sampled posterior, while the datasets 4 through 8 have at least 1000 trees in an equivalent sampled posterior, and two datasets have > 10,000).

If we set aside DS2 and DS3, the results in Zhang and Matsen as well as in the current appendix are clear that the richer models are better. The manuscript describes this accurately, saying "On the real datasets DS1 and DS4, the richer models of CCD1 and CCD2 can capture the structure of the posterior better than a CCD0 even for small number of sampled trees."

However, I'm confused by the text that follows that sentence:

"However, for non-trivial problems, sampling enough trees with MCMC to reach the regimes of CCD1, CCD2, or sample distributions is often not feasible."

This seems surprising because the advantage of the more complex models is evident with only 30 samples for the data sets with a reasonably wide posterior distribution. That observation seems at odds with the overall conclusion:

"Hence, CCD0 offers the overall best posterior estimate for hard problems (in terms of entropy). This raises the question of how to select the best model for real datasets, in particular, when they are relatively simple (in terms of entropy)."

as well as this sentence in the Discussion:

"First, CCD0 is best for few samples in terms of accuracy, precision, and stability, then CCD1 catches up and becomes the best method in the mid range, while CCD2 and the sample distribution require a huge number of sampled trees to become competitive."

Please help me understand how these statements are not contradictory.

In general, I continue to feel that a diverse collection of real datasets (using all of the DS data sets as a minimum, but I'm sure these authors have other interesting data sets at hand) is the right way to answer the question of the appropriateness of the tree model because at the end of the day we care about performance on real data. If the authors agree, I think it would make sense to expand the real data analysis and put it in the main text, rather than in an appendix.

2.

I do understand that for another part of the paper, namely the part that uses these models to find a consensus tree, it's convenient to have a known ground truth tree, motivating the use of simulated data. However, even for that application, it seems like one could use statistics such as the expected distance to the trees in the golden run posterior as a metric. I would hesitate to make such sweeping conclusions as the authors do in the conclusion without more investigation into real data sets, with their quirks and difficulties.

3.

The authors state "CCD2 which is equivalent to their sDAG model." It is true that the structure of CCD2 is equivalent to the sDAG model, but the inferential methods differ. Zhang and Matsen (2018) show an improvement to inference when adding regularization, especially when data is sparse. At least this should be acknowledged.

4.

It would seem important to provide the "denominator" of Table 1, namely the number of trees contained in the CCD. Yes, as we get more samples we are more likely to get the true tree, but also the number of expressible trees is going up very quickly. Sorry to miss this on the first round.

5.

Can the authors clarify the motivation for using AIC in this context? It seems like an appropriate choice for settings where on can't have a true out-of-sample data set, but in this case one can generate additional data set just by running the chain again. I could be missing something about the intent here.

Details:

I'm not sure if I agree with the idea that entropy quantifies whether a problem is "hard". I think that "hard" cases are ones with complex multimodal distributions. In contrast, the uniform distribution on trees would have maximum entropy but is very easy for MCMC to traverse and CCD0 would capture it well. Perhaps "diffuse" would be a better term?

Figure 3 is nice, though it would be worth explaining the visual notation you are using (sister clade) in the caption.

**Have the authors made all data and (if applicable) computational code underlying the findings in their manuscript fully available?**

Reviewer #1: Yes

Reviewer #2: **No: **The data link provided does not describe including the XML files for the real data sets. If that's the case then please amend the description.

PLOS authors have the option to publish the peer review history of their article (what does this mean?). If published, this will include your full peer review and any attached files.

Reviewer #1: No

Reviewer #2: No
---

## [Decision Letter · Decision Letter 2]

27 Nov 2024

PCOMPBIOL-D-24-00309R2

Accurate Bayesian phylogenetic point estimation using a tree distribution parameterized by clade probabilities

PLOS Computational Biology

Dear Dr. Berling,

Thank you for submitting your manuscript to PLOS Computational Biology. After careful consideration, we feel that it has merit but does not fully meet PLOS Computational Biology's publication criteria as it currently stands. Therefore, we invite you to submit a revised version of the manuscript that addresses the points raised during the review process.

Please submit your revised manuscript within 30 days Jan 27 2025 11:59PM. If you will need more time than this to complete your revisions, please reply to this message or contact the journal office at ploscompbiol@plos.org. Please include the following items when submitting your revised manuscript:

We look forward to receiving your revised manuscript.

Kind regards,

Natalia L. Komarova

Section Editor

PLOS Computational Biology

Natalia Komarova

Section Editor

PLOS Computational Biology

Feilim Mac Gabhann

Editor-in-Chief

PLOS Computational Biology

Jason Papin

Editor-in-Chief

PLOS Computational Biology

**Journal Requirements:**

1) The file inventory includes files for Figures 1a, 1b, 1c, 4a, 4b, 5a, and 5b. We would recommend either combining these into a single Figure 1.tiff file with separate internal panels, or renumbering them as individual figures, as we are not able to publish multiple components of a single figure as separate files.

2) Please ensure that the figures are uploaded in a numerical order.

3) Please label the supplementary figures and tables as figures S1, S2, etc and table S1 and amend their references in the manuscript accordingly.

**Reviewers' comments:**

Reviewer's Responses to Questions

Reviewer #2: Thank you to the authors for their revision and for continuing a constructive dialog about the manuscript. The challenges with these sorts of comparisons are not trivial.

In their response the authors justify not focusing more on real data experiments by stating:

1. the true posterior distribution is required for evaluating CCD models

2. evaluation on real data sets is impossible because it takes too long get accurate posterior tree estimates

For the first point, the key fact to keep in mind is that the true underlying posterior is not available for simulated data, just like it's not available for real data. There is no difference there. Thus, this is not itself an argument for using simulated data sets.

For the second point, there are two factors that determine the amount of time required to get a posterior sample: the number of taxa, and the shape of the posterior distribution. The number of taxa can be modulated for a real data set by subsetting, just like one can simulate data sets of different sizes.

If the shape of the posterior distribution differs between real and simulated data sets, this is a strong argument for including real data sets in the evaluation. The methods provided here are to mathematically describe the shape of the posterior, so if real data is different than simulated data then we will be misled by choosing our methods using simulated data. As a metaphor in a simpler 1-D case, one might imagine that simulated data always looks Normal, whereas real data would take a variety of shapes.

As far as I can tell, the authors do not show that the shape of the posterior distributions for simulations are similar to the shape of posterior distributions for real data. It seems to me that the one available point of comparison we have, namely the relative ranking of the models, differs significantly between simulations and the real data analysis. This seems important, even if the authors dismiss their real data analysis as trivial.

There is nothing wrong with doing a simulation study! These can be quite valuable. However, in this context it seems appropriate to make it clear in statements of conclusion that there are signs in this paper and others that conclusions would differ for real data.

**Have the authors made all data and (if applicable) computational code underlying the findings in their manuscript fully available?**

Reviewer #2: None

PLOS authors have the option to publish the peer review history of their article (what does this mean?). If published, this will include your full peer review and any attached files.

Reviewer #2: No

**Figure resubmission:**
---

## [Editor Report · Decision Letter 3]

13 Jan 2025

Dear Mr. Berling,

We are pleased to inform you that your manuscript 'Accurate Bayesian phylogenetic point estimation using a tree distribution parameterized by clade probabilities' has been provisionally accepted for publication in PLOS Computational Biology.

Best regards,

Natalia L. Komarova

Section Editor

PLOS Computational Biology

Natalia Komarova

Section Editor

PLOS Computational Biology

---

## [Editor Report · Acceptance letter]

PCOMPBIOL-D-24-00309R3

Accurate Bayesian phylogenetic point estimation using a tree distribution parameterized by clade probabilities

Dear Dr Berling,

I am pleased to inform you that your manuscript has been formally accepted for publication in PLOS Computational Biology. Your manuscript is now with our production department and you will be notified of the publication date in due course.

With kind regards,

Anita Estes
